# L-NAME Administration Enhances Diabetic Kidney Disease Development in an STZ/NAD Rat Model

**DOI:** 10.3390/ijms222312767

**Published:** 2021-11-25

**Authors:** Raphaëlle Corremans, Patrick C. D’Haese, Benjamin A. Vervaet, Anja Verhulst

**Affiliations:** Laboratory of Pathophysiology, Department of Biomedical Sciences, University of Antwerp, 2610 Wilrijk, Belgium; raphaelle.corremans@uantwerpen.be (R.C.); patrick.dhaese@uantwerpen.be (P.C.D.); benjamin.vervaet@uantwerpen.be (B.A.V.)

**Keywords:** diabetic kidney disease, preclinical model, hyperfiltration, albuminuria, creatinine, glomerular filtration rate, histopathology

## Abstract

One of the most important risk factors for developing chronic kidney disease (CKD) is diabetes. To assess the safety and efficacy of potential drug candidates, reliable animal models that mimic human diseases are crucial. However, a suitable model of diabetic kidney disease (DKD) is currently not available. The aim of this study is to develop a rat model of DKD by combining streptozotocin and nicotinamide (STZ/NAD) with oral N(ω)-Nitro-L-Arginine Methyl Ester (L-NAME) administration. Diabetes was induced in male Wistar rats by intravenous injection of 65 mg/kg STZ, 15 min after intraperitoneal injection of 230 mg/kg NAD. Rats were assigned to different groups receiving L-NAME (100 mg/kg/day) (STZ/NAD/L-NAME) or vehicle (STZ/NAD) for a period of 9 or 12 weeks by daily oral gavage. All rats developed hyperglycemia. Hyperfiltration was observed at the start of the study, whereas increased serum creatinine, albumin-to-creatinine ratio, and evolving hypofiltration were detected at the end of the study. Daily L-NAME administration caused a rapid rise in blood pressure. Histopathological evaluation revealed heterogeneous renal injury patterns, which were most severe in the STZ/NAD/L-NAME rats. L-NAME-induced NO-deficiency in STZ/NAD-induced diabetic rats leads to multiple characteristic features of human DKD and may represent a novel rat model of DKD.

## 1. Introduction

Diabetic kidney disease (DKD), also called diabetic nephropathy, is a major microvascular diabetic complication that develops in 20% to 40% of all diabetic patients, and it has become the most common cause of chronic kidney disease (CKD) leading to end-stage renal disease (ESRD) [1,2]. DKD is a clinical syndrome characterized by persistent albuminuria, decreased glomerular filtration rate (GFR), and renal histopathological decay in the presence of diabetes [3,4]. Pathophysiological changes in the kidney, responsible for the development and progression of DKD, are caused by a long-standing disturbed glucose homeostasis, which results in hemodynamic, metabolic, and structural alterations. Notwithstanding the rapidly increasing prevalence of diabetes and, consequently, DKD, a rodent model that develops the clinical and pathological features of human DKD within an acceptable experimental time frame is currently not available [5,6].

Streptozotocin (STZ), an agent toxic to pancreatic β-cells, results in absolute insulin deficiency and is widely used for the induction of diabetes in rodents. However, insulin dependency and the absence of progressive renal failure make this model unsuitable for testing new therapies to prevent or halt the progression of DKD [5]. Furthermore, it has been reported that the co-administration of nicotinamide (NAD) with STZ in rats partially rescues pancreatic β-cells from STZ-induced toxicity [7]. This results in rats with insulinopenia and high blood glucose that are not insulin dependent and that can, therefore, be followed-up for a longer period, thus providing an acceptable experimental timeframe during which the animals develop the renal pathology of diabetes in parallel with the metabolic pathology.

The progression of kidney disease in experimental models of diabetes is associated with renal hyperfiltration [8]. High intraglomerular pressure, and thus glomerular hyperfiltration, is one of the first pathological events that occur in DKD due to afferent arteriolar vasodilation and efferent vasoconstriction [9]. Endothelial dysfunction is characterized by nitric oxide (NO) deficiency, which contributes to hypertension, since NO plays a major role in regulating blood pressure [10]. In endothelial cells, endothelial nitric oxide synthase (eNOS) produces most of the vascular NO. Therefore, reduced eNOS activity is known to be responsible for reduced vessel dilatation, increased vessel stiffness, and hypertension. In humans, endothelial dysfunction and hypertension are strongly linked with diabetic nephropathy, and in mice, it is known that genetic knockouts of eNOS are more prone to developing diabetic nephropathy [11,12,13]. This prompted us to use N(ω)-Nitro-L-Arginine Methyl Ester (L-NAME), a structural L-arginine analogue, which is a widely used inhibitor of NOS and NO bioavailability and thus contributes to the progression of DKD.

We hypothesize that L-NAME-induced NO-deficiency in diabetic rats may contribute significantly to the development of all desired characteristic features of human DKD. It is, therefore, the aim of this study to develop a rat model of DKD by combining STZ/NAD with oral L-NAME administration.

## 2. Results

### 2.1. Animal Model of DKD: General Animal Aspects

The body weights of the STZ/NAD diabetic rats did not differ at the end of the study as compared to the start of the study, whereas the body weights of the control rats significantly increased from week 3 onwards until the end of the study at week 9 or week 12 (*p* < 0.05). On the other hand, the body weights of the STD/NAD/L-NAME (9 w) diabetic rats decreased significantly at the end of the study compared to their baseline values (*p* < 0.05) (Figure 1). Water intake, food consumption, and urinary volume significantly increased in all diabetic groups from week 3 onwards until the end of the study as compared to the control rats and the baseline values (Figure 1). Mortality was limited to three animals, spread over the STZ/NAD/L-NAME groups, that died before the planned sacrifice.

### 2.2. Both STZ/NAD and STZ/NAD/L-NAME Cause Glucose Intolerance in Rats

Hyperglycemia, the key feature of diabetes, was noticed from week 3 onwards until the end of the study in all diabetic rats, as compared to their baseline values and the control rats (Figure 2).

The blood glucose concentrations before (0) and 30, 90, and 180 min after oral glucose administration, at the end of the study, are shown in Table 1. No statistically significant differences were seen between the diabetic rats (STZ/NAD vs. STZ/NAD/L-NAME). However, compared to the control animals, the blood glucose levels were significantly increased at every time point, suggesting glucose intolerance in all diabetic rats.

### 2.3. Diabetes Induction Affects Kidney Function

To assess renal function, serum creatinine and creatinine clearance, as measures of GFR, were determined (Figure 3). Already at week 3, serum creatinine levels were significantly higher in the diabetic rats (both STZ/NAD and STZ/NAD/L-NAME) than in the control animals. Furthermore, in the STZ/NAD/L-NAME groups, the increase in serum creatinine appeared to be more pronounced at earlier time points: significantly increased serum creatinine levels compared to the STZ/NAD rats after 6 weeks in the group receiving L-NAME for 9 weeks and after 9 weeks in the group receiving L-NAME for 12 weeks (Figure 3A,B). Prior to the loss of renal function and, thus, reduced GFR (creatinine clearance), diabetic nephropathy is characterized by renal hyperfiltration. This was reflected by significantly increased creatinine clearance at week 3 in all diabetic animals. However, at week 9, the GFR had returned to baseline values, suggesting that, at a particular time point, the compensatory mechanisms of the remaining nephrons were no longer sufficient to maintain the elevated GFR (Figure 3C,D). The urinary albumin-to-creatinine ratio increased rapidly to an average of 93 mg/g three weeks after diabetes induction, which was five-fold higher than their baseline values. This increase was sustained in all diabetic rats until the end of the study compared to their baseline values and the control rats (Figure 3E,F).

### 2.4. L-NAME-Induced NO-Deficiency Leads to Increased Blood Pressure

Hypertension is a major risk factor in the onset and progression of DKD [14]. Elevations in blood pressure are directly correlated with the worsening of kidney disease progression in patients with diabetes [15]. In our study, daily oral administration of L-NAME significantly increased both systolic and diastolic blood pressure at week 3. However, L-NAME appeared to lose its effect as the study progressed, since statistically significant differences were no longer seen at the end of the study compared to the STZ/NAD animals (Figure 4).

### 2.5. L-NAME Administration Aggravates Renal Histopathological Changes

The tubulointerstitial lesions observed in the diabetic STZ/NAD/L-NAME animals were characterized by ruffling and thickening of the tubular basement membrane, focal fibrotic regions, tubular epithelial cell necrosis, tubular atrophy, and tubular dilatation (Figure 5). This phenotype was also present in the STZ/NAD animals, but in a noticeably milder form (Figure 5). This became clear after semi-quantitative scoring of the interstitial lesions, i.e., ruffling/thickening of the tubular basement membrane and fibrosis. Half of the STZ/NAD diabetic animals still, at the end of the study, showed normal interstitium (degree 1), while the other half showed moderate lesions (degree 2 and 3). On the other hand, none of the STZ/NAD/L-NAME animals were categorized as having normal interstitia, and 40% of them showed extensive interstitial deterioration (degree 4, <50% of normal interstitium) (Figure 6). On the other hand, tubular injury, quantified as the percentage of tubular lesions (tubules showing cell necrosis/cell loss/atrophy), did not differ significantly in the STZ/NAD and the STZ/NAD/L-NAME animals (Figure 5).

The observed glomerular damage was evaluated on Jones methenamine silver-stained sections and consisted of mild glomerular mesangial expansion and diffuse thickening of the capillary walls. This staining also allowed visualization of tubular lesions (Figure 7). The glomerular damage seems to be equally severe in the STZ/NAD as in the STZ/NAD/L-NAME diabetic rats. However, a clear example of afferent/efferent arteriole hyalinization, which is a typical feature of DKD, was found in an STZ/NAD/L-NAME animal (Figure 8).

## 3. Discussion

A crucial step in drug development is preclinical testing. To understand the physiology and pathophysiology of diseases and to assess the safety and efficacy of potential drug candidates, reliable animal models that closely mimic human diseases are crucial. However, to date, no preclinical model exhibits the functional, structural, and molecular pathological features of progressive human DKD [5,16]. Given the importance of hypertension in the early stages of diabetes development, we therefore aimed to develop a rat model of DKD by combining co-administration of STZ/NAD, to induce type 2 diabetes, with daily L-NAME administration to induce additional endothelial dysfunction. By doing so, we additionally aimed to obtain a model that is insulin-independent and that can be followed-up for a longer time, thus (i) providing an acceptable experimental time frame during which the animals could develop the renal pathology of diabetes in parallel with the metabolic pathology; and (ii) allowing intervention studies with widely varying treatment periods.

Glucose tolerance was clearly affected by the induction of diabetes. All rats developed hyperglycemia after STZ/NAD co-administration. Furthermore, clinical signs became almost immediately apparent such as excessive water consumption and excessive urination. The development and progression of DKD consists of hemodynamic, metabolic, and structural alterations. Early-phase DKD is characterized by an increase in GFR, which gradually progresses to hypofiltration over time [1]. Hyperfiltration precedes the onset of albuminuria and glomerular damage, which become clinically detectable [1,2]. Progression to overt nephropathy and further progression to ESRD requiring renal replacement therapy are associated with the duration of untreated hyperglycemia and increased blood pressure [17]. At the end of the study serum creatinine values were increased, whereas the GFR, after increasing significantly at the start of the study, had returned to baseline values. This effect on GFR is well described in DKD and is most likely due to compensatory mechanisms (i.e., hyperfiltration in the remaining nephrons) that mask underlying nephron loss. These compensatory mechanisms are eventually and gradually outweighed by persistent loss of renal mass, which may have balanced each other out at the endpoint of our study. A longer experimental period could, therefore, have revealed progression to hypofiltration [1,18]. The pathogenesis of hyperglycemia-induced glomerular hyperfiltration is complex and not fully understood. However, inactivation of the tubuloglomerular feedback and activation of the renin–angiotensin–aldosterone system (RAAS) appear to be major mechanisms for glomerular hyperfiltration in early diabetes [19]. The tubuloglomerular feedback mechanism is important in autoregulating and maintaining renal blood flow. It controls glomerular hemodynamics by constricting the glomerular afferent arteriole and thus decreases the single-nephron GFR when the macula densa cells sense an increased sodium concentration in the distal tubule [20]. However, high glucose levels in the glomerular ultrafiltrate enhance sodium-glucose cotransporter-2 (SGLT2)-mediated sodium-glucose reabsorption, which decreases sodium delivery to the macula densa and inhibits the tubuloglomerular feedback-induced vasoconstriction of the afferent arteriole. Hyperglycemia has, therefore, a prominent role in the onset of hyperfiltration, while SGLT2-inhibitors have become important therapeutic tools to prevent progression towards CKD in diabetic individuals [1,21,22,23]. Furthermore, high glucose levels also cause vasoconstriction of the efferent arteriole due to an overstimulation of the RAAS system [24]. Indeed, renin is secreted by juxtaglomerular cells in response to decreased sodium delivery to macula densa cells. Increased intrarenal angiotensin II synthesis, a potent vasoconstrictor of the efferent arteriole, further contributes to hyperfiltration [25]. Therefore, blockade of the RAAS system can also attenuate progression of DKD [26].

Albuminuria, measured as the urinary albumin-to-creatinine ratio, is an important marker for predicting the risk of ESRD in diabetes [1]. Albuminuria was increased in all diabetic rats from week 3 onwards until the end of the study. Increased glomerular permeability allows albumin to pass through the glomerular filtration barrier, which requires ultrastructural changes and indicates glomerular damage [27].

Diabetes and hypertension constitute the most notorious combination for development of DKD, which led us to administer L-NAME to diabetic animals. L-NAME, an NOS inhibitor, is associated with reduced endothelium-dependent vasodilation, eventually leading to hypertension [28]. In our study, chronic administration of L-NAME caused a rapid rise in blood pressure, but it seemed to be less effective afterwards. Contrary to our results, Li et al. concluded that increasing the exposure time to L-NAME (8 weeks, 40 mg/kg, p.o.) resulted in a greater increase in blood pressure [29]. Furthermore, Paulis et al. investigated the restoration of blood pressure after cessation of L-NAME administration. The impaired NO signaling, after 5 weeks of L-NAME administration (40 mg/kg, drinking water), was fully restored 3 weeks later [30]. In contrast, Lui et al. demonstrated that prolonged application of L-NAME contributed to nitrodilator-mediated vasodilation by releasing NO from L-NAME’s guanidino nitro group via Fenton chemistry [31]. Nevertheless, the results of the present study indicate that L-NAME administration played a role in the aggravation of the histopathological damage of the kidney.

At the functional level, L-NAME seems to induce an earlier increase in serum creatinine levels. This difference, however, did not exist anymore at the end of the study and was not accompanied by more severe hyperfiltration or albuminuria. L-NAME-induced aggravation of diabetic nephropathy was, therefore, most obvious at the histopathological level, i.e., as more severe interstitial lesions.

In general, histopathological evaluation of diabetic nephropathy revealed patterns of renal injury with mild glomerular damage relative to the severity of tubulointerstitial lesions (tubular basement membrane thickening, focal fibrosis, tubular necrosis, atrophy, and dilatation), which became even more pronounced in the STZ/NAD/L-NAME rats. The fact that the terminal stage of glomerulosclerosis, as seen in human DKD, seems not to have been reached can be considered the major limitation of the study. On the other hand, our results are in accordance with those of Di Vincenzo et al., who focused on the renal structural changes in type 2 diabetes. They stated that renal lesions in type 2 diabetes are heterogeneous, and that diabetic glomerulopathy is not a predominant feature, but that tubulointerstitial and vascular abnormalities occur more regularly [32]. Additionally, Fioretto et al. described the discrepancy between tubular and glomerular renal lesions, leading to renal dysfunction in type 1 and type 2 diabetes [33]. Our model, being a type 2 rather than type 1 diabetes-induced DKD model, corroborates the fact that our animals remained insulin-independent throughout the study.

In conclusion, a suitable rodent model featuring the functional/histopathological characteristics of DKD is of utmost importance to examine the mechanisms of diabetic renal injury and to evaluate potential therapeutic interventions. The latter is important in both the early (perhaps reversible) phase of diabetes and in established and irreversible DKD. We preferred to first develop a rat model with diabetic nephropathy, since (1) in general, rats are more prone to develop CKD compared to mice, and (2) rats allow easier collection and larger volumes of blood and urine samples. Our STZ/NAD rat model exhibits hyperglycemia-induced hyperfiltration, increased serum creatinine, albuminuria, as well as tubulointerstitial damage and early glomerular lesions. Additionally, L-NAME administration increased both systolic and diastolic blood pressure, and it aggravated renal interstitial structural damage. Although the terminal stage of glomerulosclerosis was not reached in the current study period, the renal injury pattern already demonstrated mild glomerular lesions, relative to the more severe tubulointerstitial damage. Hence, we propose the STZ/NAD/L-NAME model as a relevant type 2 diabetes-induced DKD model.

## 4. Materials and Methods

### 4.1. Drugs

STZ (Sigma-Aldrich, St. Louis, MO, USA) and NAD (Sigma-Aldrich, St. Louis, MO, USA) were dissolved in physiological water at concentrations of 65 mg/mL and 115 mg/mL, respectively, before intravenous (i.v.) and intraperitoneal (i.p.) injection, respectively. L-NAME (Sigma-Aldrich, St. Louis, MO, USA) was dissolved in water at a concentration of 10 mg/mL before daily oral gavage. A dose volume of 10 mL/kg was used.

### 4.2. Study Design

Fifty-two (52) male Wistar rats (Iffa, Credo, Brussels, Belgium) were housed two per cage, exposed to a 12–12 h light–dark cycle, with free access to water and their allotted diet. Diabetes was induced chemically by administration of an i.v. injection of 65 mg/kg STZ 15 min after i.p. injection of 230 mg/kg NAD. When blood glucose levels were <200 mg/dL, a second injection of both compounds was given one week later. Diabetes rats were randomly assigned to 4 different groups to receive either L-NAME or vehicle for a period of 9 or 12 weeks. As a study reference, a group of non-diabetic control rats with normal renal function was also included (Figure 9).

Every 3 weeks, the animals were housed in metabolic cages for 24 h to collect urine samples followed by blood sampling through the tail vein in a restrained, conscious condition. The animals were sacrificed by exsanguination through the retro-orbital plexus after anesthesia by i.p. administration of 80 mg/kg ketamine (Ketalar^®^; Pfizer, Brussels, Belgium) and 16 mg/kg xylazine (Rompun^®^; Bayer, Diegem, Belgium).

### 4.3. Blood Pressure

Blood pressure was measured at baseline and every 3 weeks until sacrifice, non-invasively in restrained awake rats using a tail-cuff system (CODA system, Kent Scientific, Torrington, CT, USA). The rats were trained before the actual measurements to reduce stress and variability during the procedure.

### 4.4. Biochemical Analyses

To assess renal function, creatinine was measured in serum and urine samples according to the Jaffe principle. Urinary albumin concentration was measured using Hemocue^®^ Albumin 201 System (HemoCue AB, Ängelholm, Sweden). Blood glucose levels were determined every 3 weeks (GlucoMen Lx Plus+, A. Menarini diagnostics, Machelen, Belgium).

### 4.5. Oral Glucose Tolerance Test (OGTT)

The OGTT was performed in overnight fasted rats every 3 weeks until sacrifice. Blood glucose levels were measured before (T0), and 30 min, 90 min, and 180 min after oral administration of a glucose solution (2 g/kg body weight).

### 4.6. Histology

The left kidney was isolated and decapsulated, after which a transverse slice was fixed (4 h) in neutral buffered formalin, rinsed with isopropanol, and embedded in paraffin (52 °C; Kendall, Mansfield, MA, USA). General morphological evaluation was performed on 4 µm-thick sections that were stained with Periodic acid-Schiff (PAS) or Jones Methenamine Silver-stain. Quantification of the percentage of tubular lesions and the severity of interstitial lesions was performed on the PAS-stained sections. Four random pictures (200×) were taken of the cortex of each PAS-stained renal section. Using Fiji image analysis software (US National Institutes of Health, Bethesda, MD, USA), the area that contained injured (cell death/cell loss/atrophy) tubules was measured. The ratio of the injured tubules area to the total area resulted in the percentage of area taken by the damaged tubules. Interstitial lesions, i.e., ruffling and thickening of the tubular basement membrane and focal fibrosis, were quantified semi-quantitatively in a blinded manner. The lesions were scored using four different categories: degree 1 (normal interstitium), degree 2 (>70% normal interstitium), degree 3 (>50% normal interstitium), and degree 4 (<50% normal interstitium).

### 4.7. Statistics

The results are expressed as means ± SEM. Non-parametric statistical analyses were performed using Prism (GraphPad Software, San Diego, CA, USA). A *p*-value of <0.05 was considered statistically significant. Semi-quantitative scoring was statistically evaluated using the Chi-square test.

## Figures and Tables

**Figure 1 ijms-22-12767-f001:**
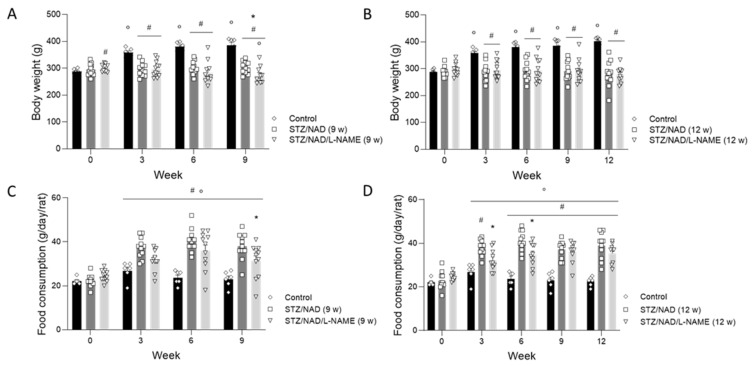
Changes in body weight (**A**,**B**), food consumption (**C**,**D**), water intake (**E**,**F**), and urinary volume (**G**,**H**) in STZ/NAD and STZ/NAD/L-NAME rats for a period of 9 or 12 weeks. Data are presented as mean ± SEM. * *p* < 0.05 vs. STZ/NAD, # *p* < 0.05 vs. non-diabetic control animals, and ° *p* < 0.05 vs. baseline values.

**Figure 2 ijms-22-12767-f002:**
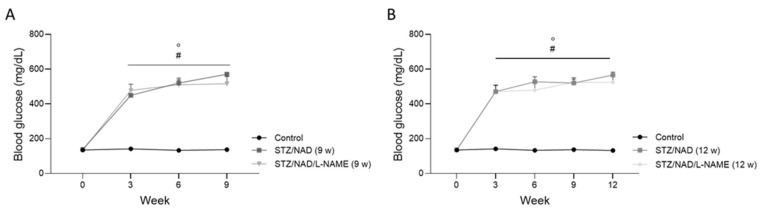
Blood glucose levels in diabetic STZ/NAD and STZ/NAD/L-NAME rats for a period of 9 (**A**) or 12 weeks (**B**) and non-diabetic control rats. Data are presented as mean ± SEM. * *p* < 0.05 vs. STZ/NAD, # *p* < 0.05 vs. non-diabetic control animals, and ° *p* < 0.05 vs. baseline values.

**Figure 3 ijms-22-12767-f003:**
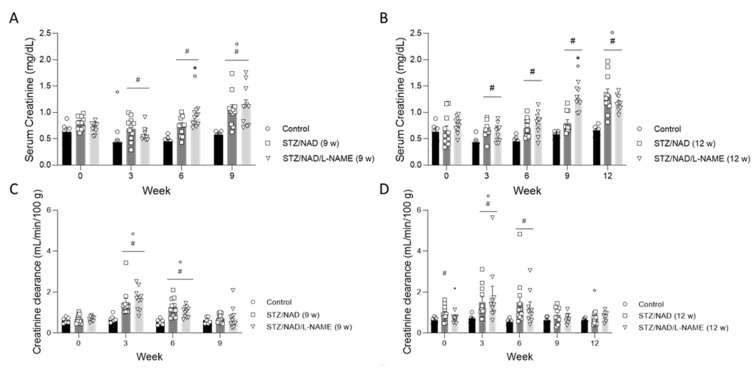
Serum creatinine (**A**,**B**), creatinine clearance (**C**,**D**), and urinary albumin-to-creatinine ratio (**E**,**F**) in STZ/NAD- and STZ/NAD/L-NAME rats for a period of 9 or 12 weeks. Data are presented as mean ± SEM. * *p* < 0.05 vs. STZ/NAD, ** *p* < 0.01 vs. STZ/NAD, # *p* < 0.05 vs. control, and ° *p* < 0.05 vs. week 0.

**Figure 4 ijms-22-12767-f004:**
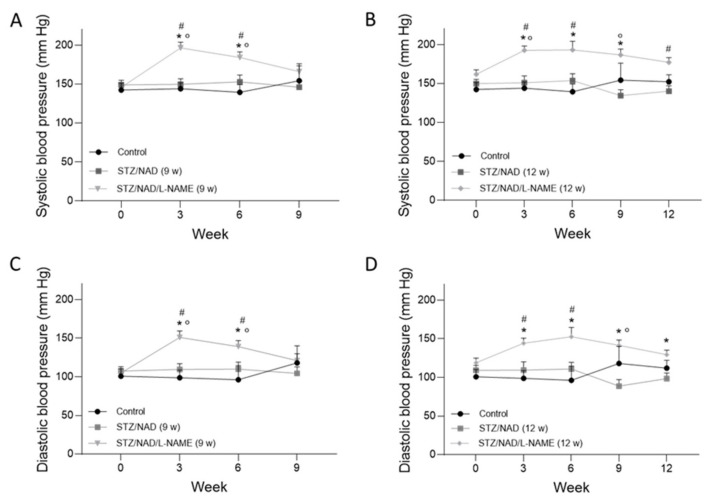
Systolic (**A**,**B**) and diastolic (**C**,**D**) blood pressure of STZ/NAD and STZ/NAD/L-NAME diabetic rats for a period of 9 (**A**,**C**) or 12 weeks (**B**,**D**) and non-diabetic control rats. Data are presented as mean ± SEM. * *p* < 0.05 vs. STZ/NAD, # *p* < 0.05 vs. control, and ° *p* < 0.05 vs. week 0.

**Figure 5 ijms-22-12767-f005:**
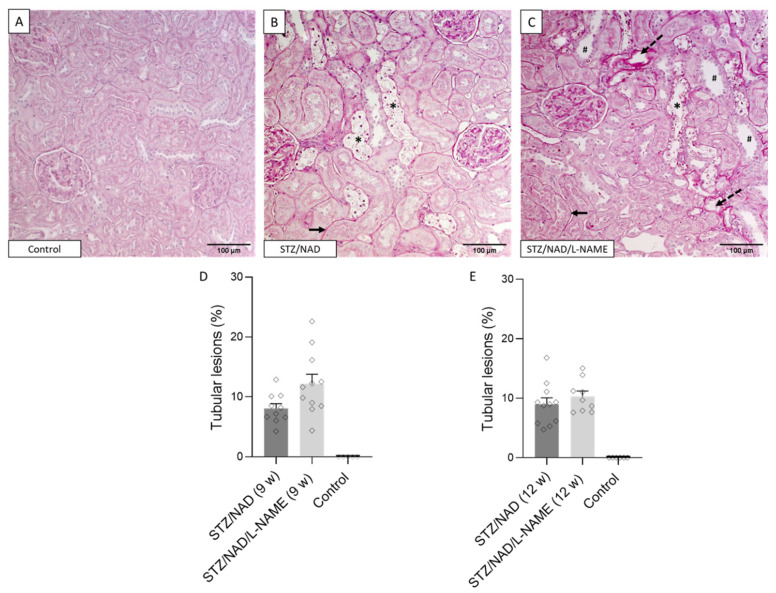
Histopathology staining of DKD. PAS-stained renal sections of a non-diabetic control rat (**A**), STZ/NAD (**B**) and STZ/NAD/L-NAME diabetic rat at week 12 (**C**). Tubular epithelial cell necrosis (asterisk), tubular atrophy (dashed arrow), thickening of tubular basement membrane (arrow), and tubular dilatation (#). Tubular lesions were quantified in control, STZ/NAD, and STZ/NAD/L-NAME rats after 9 weeks (**D**) and 12 weeks (**E**).

**Figure 6 ijms-22-12767-f006:**
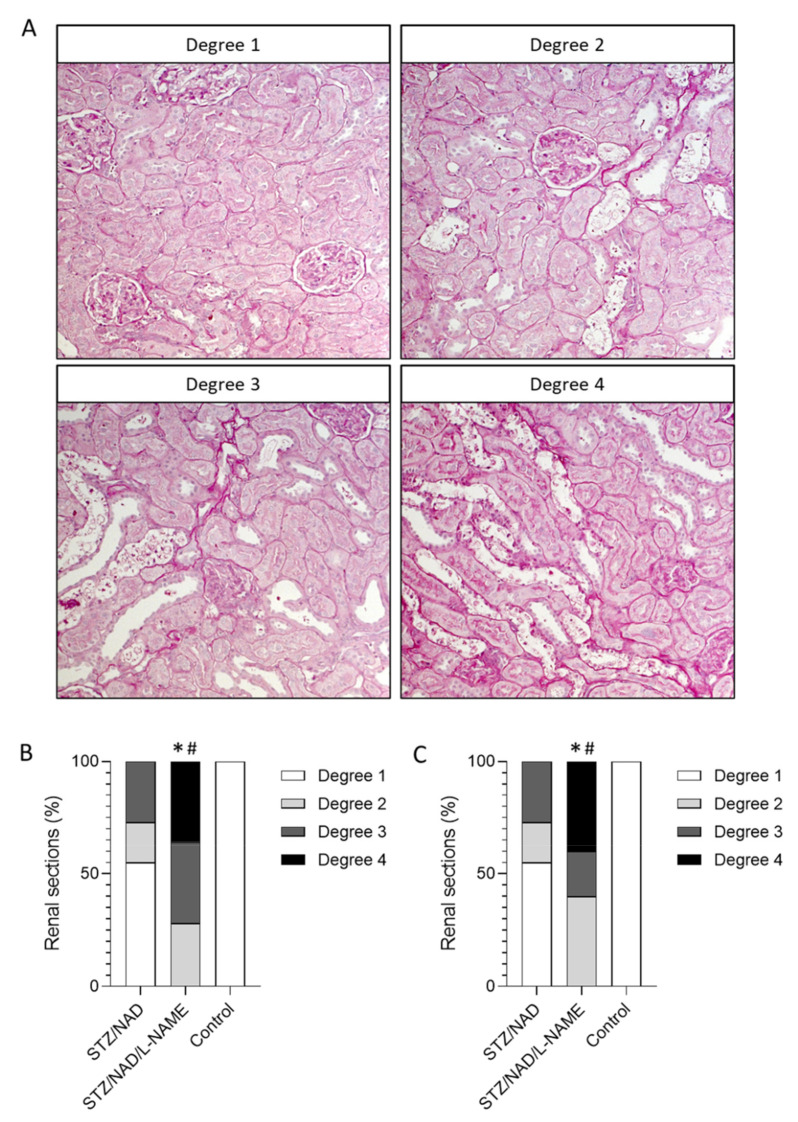
Semi-quantitative grading of renal interstitial histopathology according to 4 different degrees of severity (**A**). PAS-stained renal sections of STZ/NAD, STZ/NAD/-L-NAME diabetic rats and non-diabetic control rats were analyzed blinded by a researcher at week 9 (**B**) and week 12 (**C**) and divided into the following categories: Degree 1: normal interstitium; degree 2: >70 % normal interstitium; degree 3: >50% normal interstitium; degree 4: <50% normal interstitium. Chi-square analysis was performed. * *p* < 0.05 vs. STZ/NAD, # *p* < 0.05 vs. control.

**Figure 7 ijms-22-12767-f007:**
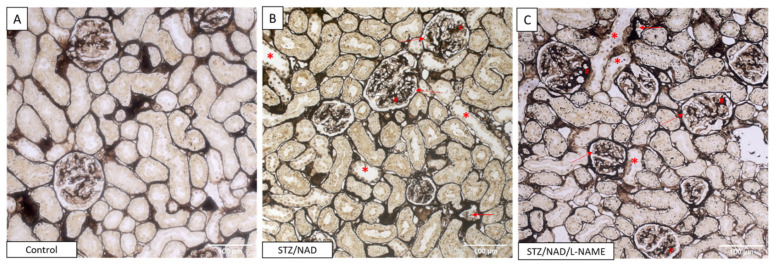
Jones methenamine silver-stained renal section of a non-diabetic control rat (**A**), STZ/NAD (**B**) and STZ/NAD/L-NAME (**C**) diabetic rat at week 12. Glomeruli of the diabetic rat show mild mesangial expansion (#) and diffuse thickening of capillary walls (dashed arrow). Tubular lesions manifested as tubular epithelial cell necrosis (asterisk) and tubular atrophy (arrow).

**Figure 8 ijms-22-12767-f008:**
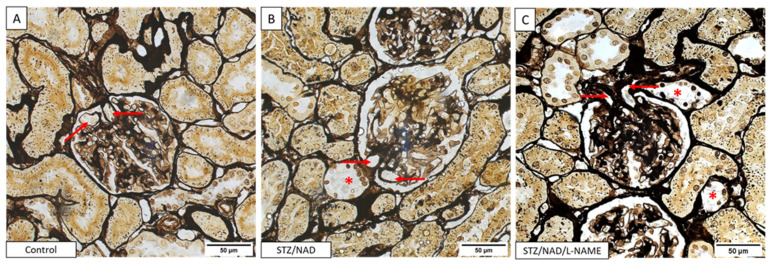
Jones methenamine silver-stained renal sections of a control (**A**), STZ/NAD (**B**), and STZ/NAD/L-NAME (**C**) rat after 12 weeks. The glomerulus exhibited hyalinosis of both the afferent and the efferent (arrow) arterioles in the STZ/NAD/L-NAME diabetic rats. Tubular epithelial cell injury (asterisk) was also visible.

**Figure 9 ijms-22-12767-f009:**
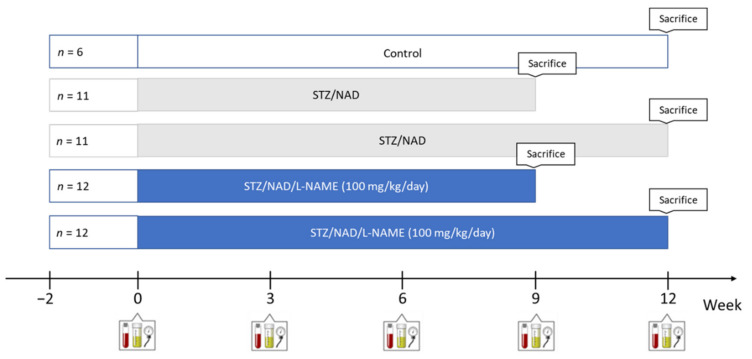
Study setup. Created with BioRender.

**Table 1 ijms-22-12767-t001:** Blood glucose concentrations during glucose tolerance test in diabetic and control rats. Glucose (2 g/kg) was administered via oral gavage (0 min), and blood glucose was measured after 30, 90, and 180 min. Data are presented as mean ± SEM.

Week	Group	Blood Glucose Concentration (mg/dL)			
		0 min	30 min	90 min	180 min
9	Control	105 ± 4	130 ± 6 °	129 ± 6 °	115 ± 9
	STZ/NAD	285 ± 40 ^#^	504 ± 24 °^#^	533 ± 22 °^#^	505 ± 17 °^#^
	STZ/NAD/L-NAME	440 ± 55 ^#^	505 ± 47 °^#^	517 ± 44 °^#^	476 ± 49 °^#^
12	Control	100 ± 4	133 ± 3 °	128 ± 5 °	113 ± 7
	STZ/NAD	369 ± 48 ^#^	499 ± 44 °^#^	514 ± 31 °^#^	505 ± 40 °^#^
	STZ/NAD/L-NAME	407 ± 49 ^#^	513 ± 34 °^#^	525 ± 35 °^#^	459 ± 46 ^#^

^#^ *p* < 0.05 versus control. ° *p* < 0.05 versus 0 min.

## Data Availability

The data presented in this study are available on request from the corresponding author.

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
