# Peer review of "L-NAME Administration Enhances Diabetic Kidney Disease Development in an STZ/NAD Rat Model"

_ijms, 2021, doi:10.3390/ijms222312767_

Round 1

Reviewer 1 Report

Subject Appropriateness of the Manuscript:

The topic of this manuscript falls within the scope of International Journal of Molecular Sciences

Recommendation

Accept for publication after major revision

Comments

In the manuscript entitled: “L-NAME administration enhances diabetic kidney disease development in a STZ/NAD rat model” the Authors aimed to establish a reliable animal model that closely mimic human diabetic chronic disease and could be useful to examine the mechanisms of diabetic renal injury or evaluate new potential therapeutic options.  For this purpose they developed  a rat model of DKD by combining streptozotocin and nicotinamide (STZ/NAD) with oral N(ω)-Nitro-L-Arginine Methyl Ester (L-NAME) administration.

The authors conclude that administration of STZ/NAD/L-NAME in rat leads to many of the hallmarks of human DKD, i.e. hyperglycemia-induced hyperfiltration, increased serum creatinine, albuminuria, and development of kidney injury with mild glomerular damage in relation to the severity of tubulointerstitial lesions, and as such, may represent a new preclinical model of DKD.

The manuscript, however, has some limitations, the most important of which are:

  1. the limitations of the study should be described at the end of the paper
  2. how serum glucose levels were determined by GlucoMen Lx Plus+ automated whole blood glucose meter?
  3. how the authors explain that glucose levels do not change between 30, 90 and 120 min. of the glucose tolerance test.
  4. authors should use ACR instead of albuminuria as a more reliable indicator of kidney damage. The KDIGO guidelines recommend a urinary albumin to creatinine ratio for assessing the severity of kidney injury and as a prognostic marker for chronic kidney disease.

Author Response

Yours sincerely, 

Reviewer 2 Report

Body weights and kidney failure parameters should be presented for each mice, since it is a fairly small group of animals. 

Author Response

Yours sincerely, 

Round 2

Reviewer 1 Report

Regarding my remark: "How serum glucose levels were determined by GlucoMen Lx Plus+ automated whole blood glucose meter?" The question concerned the type of material tested for glucose determination (whole blood instead of serum). Of course, glucose was assessed in whole blood, while the authors use the term whole blood / serum interchangeably in the manuscript text.  Have you used any conversion between whole blood glucose and serum glucose?  

Author Response

Please see the attachment for our response.

Yours sincerely, 

Raphaëlle Corremans
